# Exercise and Sestrin Mediate Speed and Lysosomal Activity in *Drosophila* by Partially Overlapping Mechanisms

**DOI:** 10.3390/cells10092479

**Published:** 2021-09-19

**Authors:** Alyson Sujkowski, Robert Wessells

**Affiliations:** Department of Physiology, School of Medicine, Wayne State University, Detroit, MI 48201, USA; asujkows@med.wayne.edu

**Keywords:** Sestrin, exercise, *Drosophila*

## Abstract

Chronic exercise is widely recognized as an important contributor to healthspan in humans and in diverse animal models. Recently, we have demonstrated that Sestrins, a family of evolutionarily conserved exercise-inducible proteins, are critical mediators of exercise benefits in flies and mice. Knockout of Sestrins prevents exercise adaptations to endurance and flight in *Drosophila*, and similarly prevents benefits to endurance and metabolism in exercising mice. In contrast, overexpression of dSestrin in muscle mimics several of the molecular and physiological adaptations characteristic of endurance exercise. Here, we extend those observations to examine the impact of dSestrin on preserving speed and increasing lysosomal activity. We find that dSestrin is a critical factor driving exercise adaptations to climbing speed, but is not absolutely required for exercise to increase lysosomal activity in *Drosophila*. The role of Sestrin in increasing speed during chronic exercise requires both the TORC2/AKT axis and the PGC1α homolog *spargel*, while dSestrin requires interactions with TORC1 to cell-autonomously increase lysosomal activity. These results highlight the conserved role of Sestrins as key factors that drive diverse physiological adaptations conferred by chronic exercise.

## 1. Introduction

As global populations age, the burden of caring for a disproportionately older demographic and their rising medical costs has become a major healthcare problem [1,2]. Interventions that can preserve healthy function at advanced ages are thus of substantial importance. Preservation of mobility is a major concern in aging populations, and declining mobility is associated with reduced independence [3]. Individuals with healthy mobility tend to experience fewer falls and retain the ability to travel to essential healthcare visits. Furthermore, preservation of mobility allows individuals to continue social interactions, increasing personal satisfaction and morale into advanced age [4,5]. Thus, identification of easily accessible, low-cost interventions that preserve healthy lifespan and mobility in the elderly would have widespread benefits, reducing healthcare costs and enhancing quality of life for aging individuals and those responsible for their care.

One effective intervention that reduces age-related functional decline in humans and animal models is endurance exercise [6]. Endurance exercise induces remodeling in muscle tissue, which triggers cell non-autonomous effects that improve physiological and metabolic health in multiple organ systems [7,8,9]. Studies in both invertebrate and vertebrate models show that endurance exercise promotes increased mitochondrial biogenesis/efficiency [10,11], decreased triglyceride storage [12,13], improved insulin sensitivity [14,15], and protection of both muscle and neural functions [16,17,18,19]. However, widespread application of endurance exercise as an anti-aging therapeutic is limited by a number of barriers. Many elderly individuals are unable to reach the level of training necessary to generate these adaptations because of advanced age, injury, illness, or lifestyle factors leading to long sedentary periods [20,21]. Therefore, the identification of conserved factors that induce the benefits of exercise, even without training, would have enormous therapeutic potential. One such conserved factor has been recently identified in the Sestrin family of proteins.

Sestrins are small stress-activated proteins conserved from invertebrates to humans [22]. Three Sestrins are tissue-specifically expressed in mammals (mSesn1–3), while *Drosophila* and *C. elegans* express single Sestrin orthologues (*dSesn* and *cSesn*, respectively) [23]. *Drosophila Sestrin* (*dSesn*) and mammalian *SESN1* are highly expressed in skeletal and cardiac muscle [23,24,25]. Sestrins have intrinsic antioxidant function [22], which can contribute to attenuation of aging [26]. However, independent of their antioxidant activity, Sestrins can modulate mTOR signaling, in part through AMPK activation [27]. Sestrin inhibits TORC1 and enhances TORC2 activity [23,28] coordinating metabolism and healthspan. Inhibition of TORC1 by Sestrins can also induce autophagy [29], essential for elimination and recycling of damaged macromolecules and organelles from cells. Studies in *Drosophila* have established dSestrin (*dSesn*) as a critical negative feedback regulator of mTOR, whose absence results in underactivation of AMPK and overactivation of mTOR. Loss of *dSesn* leads to accelerated development of several age-related pathologies including lipid accumulation, mitochondrial dysfunction, accumulation of protein aggregates, cardiac arrhythmia and muscle degeneration [23]. These pathologies, which are strikingly similar to age- and obesity-associated human diseases, were partially prevented by mTOR inhibition through AICAR, metformin or rapamycin [23].

We have recently found that Sestrins are necessary and sufficient for beneficial adaptations to muscle function and metabolism in flies and mice [30]. In *Drosophila*, *dSesn* is required for exercise-induced improvements in climbing endurance and flight performance [30]. Here, we extended our invertebrate studies to examine exercise- and dSesn-dependent changes to climbing speed and lysosomal activity. We found that in *Drosophila*, *dSesn* mediates exercise-induced improvements in climbing speed, and that this effect requires both the TORC-2/AKT axis and PGC1α. In contrast, *dSesn* is not required for chronic exercise to simulate lysosomal activity in adipose tissue. However, when *dSesn* is overexpressed in muscle, it can act additively with exercise, in a TORC1-dependent way, to further increase adipose lysosomal activity during chronic exercise. Adding further complexity, *dSesn* overexpression in the heart induced lysosomal activity cell-autonomously, whether flies were exercised or not.

## 2. Materials and Methods

### 2.1. Fly Stocks and Maintenance

Wild-type, *dSesnXP4* (*UAS-dSesn*) and *dSesn^8A11^* (*dSesn^−/−^*) were previously produced in Exelixis *w^1118^* background. *Mef2-Gal4* (#27390), *MHC-GS-Gal4* (#43641), *ap-Gal4* (#3041), *UAS-dAKT* (#8191), *UAS-dAKT*^RNAi^ (#33615), *UAS-dSin1*^RNAi^ (#32371), *UAS-4E-BP*^RNAi^ (#36667), *UAS-dTSC2*^RNAi^ (#34737), UAS-*dPGC1*^RNAi^ (#33915), UAS-*dSnf1*^RNAi^ (#32371) and fly lines with an attP landing platform were obtained from the Bloomington *Drosophila* Stock Center (BDSC, Bloomington, IN, USA). *UAS-dSesn*^RNAi-A^ (#38481) and *UAS-dSesn*^RNAi-B^ (#104365) were obtained from the Vienna *Drosophila* RNAi Center (VDRC, Vienna, Austria). *UAS-dPGC1* is a kind gift from David Walker. *dSesn* cDNA14 of wild-type (*dSesn^WT^*) and C86S(*dSesn^CS^*), D424A (*dSesn^DA^*) or D423A/D424A (*dSesn^DDAA^*) mutated forms were previously produced and are described in Kim et al., [30]. C86, D423 and D424 in *dSesn* correspond to C125, D406 and D407 in mammalian SESN2.

Flies were grown and aged on 10% sugar-yeast medium at 25 °C, 50% humidity, and 12/12 h light/dark cycle. Control flies for all non-gene-switch UAS Gal4 experiments consisted of both the *UAS* and *Gal4* lines into *Attp* or *w^1118^*, as appropriate. For experiments in which both the *Gal4* control and *UAS* control flies were found to be statistically identical, results were pooled and graphed as a single datum. In gene-switch experiments, RU-flies of the same genotype served as the negative control.

### 2.2. Exercise Training

Cohorts of at least 800 flies were collected under light CO_2_ anesthesia within 2 h of eclosion and separated into vials of 20. Flies were then further separated into 2 large cohorts of at least 400 flies, which served as exercised and unexercised groups. Exercised groups received three weeks of ramped exercise as described [31]. The unexercised groups were placed on the exercise training device at the same time as the exercised groups, but were prevented from running by the placement of a foam stopper low in the vial. Both cohorts were housed in the same incubator with normal foam stopper placement at all times other than during an exercise bout.

### 2.3. Genetic Controls

*dSesn^8A11/8A11^* (*dSesn^−/−^*) flies were compared to their *w^1118^* background [23]. For *dSesn* RNAi experiments, adult progeny were age-matched and randomly split into control (OFF) and experimental (ON) groups. Flies in this study were longitudinally assessed from the same cohorts as Kim et al. [30]. The ON group received 100 μM mifepristone (Cayman Chemical, Ann Arbor, MI), which activates the gene switch (GS) driver, while the OFF group received the same volume of vehicle solution (70% ethanol). *dSesn* RNAi efficacy was verified in the initial study, as flies originate from the same replicate biological cohorts [30]. For *dSesn* overexpression experiments, the control attP (#24486 from BDSC) strain, the background for *UAS-dSesn^WT^*, was crossed to Mef2-Gal4.

### 2.4. Climbing Speed

Negative geotaxis was assessed by Rapid Negative Geotaxis (RING) assays in groups of 100 flies as described [32]. Briefly, vials of 20 flies were tapped down 4 times each, then measured for climbing distance 2 s after the final tap. For each group of vials, an average of 4 consecutive trials was calculated and analyzed using ImageJ (Bethesda, MD). Flies were longitudinally tested 5 times per week for 4–5 weeks. Between assessments, flies were returned to food vials and housed normally as described above. Negative geotaxis results were analyzed using two-way ANOVA analysis with *post hoc* Tukey multiple comparison tests in GraphPad Prism (San Diego, CA, USA). All negative geotaxis experiments were performed in duplicate or triplicate, with one complete trial shown in each graph.

### 2.5. qRT PCR

cDNA was prepared from 20 whole flies. Three independent RNA extractions were prepared for each sample. Differences between genotypes were assessed by ANOVA. Primer sequences are listed below.

Atg8a 5′-: GCGCATCGGTGATTTGGACAAGAA

Atg8a 3′-: TGCGCTTGCGAATGAGGAAGTAGA

rp49 5′-: ACTCAATGGATACTGCCCAAGAA

rp49 3′-: CAAGGTGTCCCACTAATGCATAAT

Relative message abundance was determined by amplification and staining with SYBR Green I using an ABI 7300 Real Time PCR System (Applied Biosystems, Waltham, MA, USA). Expression of *Rp49* and corresponding control (gal4/+, UAS-RNAi/+) flies were used for normalization.

### 2.6. Lysotracker

Lysotracker staining of adult fat bodies and myocardium was performed as in [33]. Adult flies were partially dissected, ventral side up, in room temperature PBS to expose hearts and fat bodies, then rinsed once in fresh PBS. Preps were exposed to 0.01 µM of Lysotracker green (Molecular Probes, Eugene, OR, USA) in PBS for 30 s, then washed 3 times in fresh PBS. Stained tissue was mounted in Vectashield reagent (Vector Laboratories, Burlingame, CA, USA) and imaged in the Department of Physiology Confocal Microscopy Core at Wayne State University School of Medicine on a Leica DMI 6000 with a Crest X-light spinning disc confocal using a 63X oil immersion objective (W. Nuhsbaum, Inc., McHenry, IL, USA). Images were analyzed using ImageJ (NIH, Bethesda, MD, USA). A minimum of 5 samples were analyzed for each genotype and duplicate or triplicate biological cohorts were assessed for each group. Significance was assessed by ANOVA with Tukey post-hoc comparison.

## 3. Results

### 3.1. Sestrin Drives Exercise-Induced Increases to Climbing Speed and Atg8a Expression

We have established an endurance exercise paradigm for *Drosophila* that induces several conserved adaptations after 3 weeks of training [31]. Control flies retain faster climbing speed than age-matched unexercised siblings as described previously (Figure 1A). *Sesn^−/−^* flies have lower climbing speed than control flies across ages, and do not increase climbing speed with exercise (Figure 1A). In contrast, muscle specific *dSesn* expression is sufficient to increase climbing speed in both a wild-type and *dSesn*-deficient background (*Attp;Mef2 > dSesn^WT^*, *Sesn^−/−^;Mef2 > dSesn^WT^)*, whether exercised or not (Figure 1B).

Both Sestrin and endurance exercise are known to enhance autophagy [13,29], so we next examined *autophagy-specific gene 8a* (*Atg8a*) expression, a factor that controls the rate of autophagy [34], in these same flies. Exercised background control flies have higher *Atg8a* expression than age-matched, unexercised siblings (Figure 1C). *dSesn^−/−^* flies do not increase *Atg8a* expression after exercise, but flies overexpressing *dSesn* in muscle have high *Atg8a* expression whether exercised or not (Figure 1C). Taken together, these results indicate that *dSesn* is required for exercise to induce increases in speed and *Atg8a* expression, and that overexpression of *dSesn* can mimic the effect of exercise on speed and *Atg8a* expression.

### 3.2. Dsesn Induces Lysosomal Activity in Parallel with Exercise

In order to examine tissue-specific lysosomal activity, we performed Lysotracker staining in the fat body and myocardium (Figure 1F,I). Unexercised *dSesn^−/−^* flies have low Lysotracker staining in the fat body and myocardium (Figure 1D,E), but increased fat body Lysotracker staining with exercise (Figure 1D). On the other hand, muscle-specific *dSesn* overexpression increases Lysotracker in both exercised and unexercised fat bodies and myocardium. Endurance exercise and Sestrin have been observed to enhance lysosomal activity, including autophagy, in multiple muscle tissues in flies and mammals [17,29,35,36,37,38,39], and we have previously shown that exercise increases Lysotracker staining in *Drosophila* fat body [13,17,34]. Background control flies increase fat body Lysotracker staining with exercise (Figure 1H upper panels, quantification in Figure 1D), but not myocardial Lysotracker staining (Figure 1H lower panels, quantification in E). Sestrin-deficient flies have significantly reduced Lysotracker staining at baseline in both the fat body and myocardium (Figure 1D,E,H). Exercise increases lysotracker levels in *dSesn* mutants, but only to the level of wild type unexercised, not to the level of wild type exercised flies (Figure 1D,E,H). Muscle specific *dSesn* overexpression increases fat body Lysotracker staining in unexercised flies, and exercise further increases it, indicating an additive effect (Figure 1F,I). Muscle-specific *dSesn* expression has similar effects in a *dSesn^−/−^* background, meaning expression of Sestrin in muscle is sufficient to increase fat body Lysotracker, even in flies where *dSesn* is deleted genetically in all other tissues (Figure 1G,I). This tissue-autonomous effect of *dSesn* overexpression is independent of exercise, as exercise does not significantly modify Lysotracker staining in the myocardium (Figure 1E,G,H,I).

To address muscle-specific requirements for *dSesn*, we expressed two RNAi constructs against *dSesn* specifically in adult skeletal muscles (*MHC GS > dSesn*^RNAi-A^, *MHC GS > dSesn*^RNAi-B^) and assessed climbing speed and lysosomal activity with exercise. We have previously shown that *dSesn* expression in muscle is high and induced by endurance exercise [30]. Muscle-specific expression of both RNAi constructs abolished climbing speed improvement with exercise, and muscle-specific *RNAi-B* expression reduced baseline climbing speed across ages (Figure 2A,B). Similarly, both *MHC GS > dSesn*^RNAi-A^ and *MHC GS > dSesn*^RNAi-B^ flies have low fat body (Figure 2C,D) lysosomal activity at baseline. However, when exercised, muscle-specific *dSesn* RNAi flies increase fat body Lysotracker, but not as much as exercised wild-type flies, again consistent with an additive effect between exercise and *dSesn* in this context (Figure 2C,D, myocardial Lysotracker: see Appendix A). Together with the results from Figure 1, these results indicate that *dSesn* is required in muscle to improve climbing speed during chronic exercise, and that muscle-specific *dSesn* expression is sufficient to drive increased speed across ages. Further, *dSesn* acts in muscle to cell-autonomously drive Lysotracker staining in myocardium, but acts additively with exercise to cell-non-autonomously drive Lysotracker staining in the fat body.

### 3.3. Exercise-Induced Adaptations to Climbing Speed and Lysosomal Activity Require both Oxidoreductase and TOR-Modulating Functions of Sestrin

We next expressed *dSesn* constructs containing specific functional mutations in a *dSesn*-deficient background in order to determine the mechanisms by which *dSesn* improves climbing speed and modifies lysosomal activity. The C86S mutation in Sestrin disrupts oxidoreductase function, and the D423A and D424A substitutions eliminate TORC1 inhibition and TORC2/AKT activation [28]. Wild-type *dSesn* expression in muscle rescued climbing speed in *dSesn*-deficient flies (Figure 3A), but expressing *dSesn* with the C86S mutation or with D423A and D424A substitutions did not (Figure 3A,B, compare to red *dSesn^−/−^;Mef2 > dSesn^WT^* flies).

Wild-type *dSesn* rescued fat body Lysotracker staining to normal levels in *dSesn^−/−^* flies, and exercise again increased Lysotracker additively (Figure 3C). *dSesn^−/−^* flies expressing a rescue construct containing the C86S mutation had low baseline Lysotracker in the fat body, but were able to increase Lysotracker in response to chronic exercise (Figure 3C). When a rescue construct containing the D423A and D424A substitutions was expressed instead, both the effects of dSestrin and the effects of exercise on Lysotracker were blocked. Neither mutant construct significantly altered the cell-autonomous increase of Lysotracker when *dSesn* is expressed in myocardium (Figure 1E,F). Taken together, these results indicate that both the oxidoreductase and the TOR-modulating activities of *dSesn* are required for *dSesn* to tissue-non-autonomously increase Lysotracker in adipose tissue. However, the oxidoreductase activity is not required for exercise to increase Lysotracker in parallel with dSestrin.

### 3.4. AKT Is Critical for Sestrin to Improve Climbing Speed and Enhance Lysosomal Activity

We have previously shown that both Sestrin and exercise upregulate TORC2-mediated AKT phosphorylation [30]. RNAi against Akt in muscles (*Mef2 > dSesn^WT^; Mef2 > Akt*^RNAi^) completely abolished the climbing speed improvement conferred by *dSesn* overexpression (Figure 4A). In contrast, RNAi against an essential subunit of AMPK (*Mef2 > dSesn^WT^;Mef2 > SNF1A*^RNAi^) reduced, but did not completely abolish, the climbing speed improvement conferred by *dSesn* overexpression (Figure 4B). Likewise, RNAi against Akt abolished the effect of *dSesn* on fat body lysosomal activity, but knockdown of AMPK did not block the effects of *dSesn* overexpression (Figure 4C,D,G,H). Myocardial Lysotracker was not induced by *dSesn* in the presence of RNAi against Akt (Figure 4E,F, rightmost panels in G,H). These results indicate that AMPKα in muscles is only partially required for the climbing speed enhancing effects of Sestrin, while Akt is absolutely necessary for adaptations to both climbing speed and lysosomal activity.

### 3.5. The TORC1 Axis Is Dispensable for Sestrin’s Effects on Climbing Speed and Lysosomal Activity

Sestrin is known to inhibit TORC1 in addition to potentiating TORC2 activity [23,28]. Muscle specific *dSesn* expression rescues climbing speed in flies expressing RNAi against TSC2 (*Mef2 > dSesn^WT^;Mef2 > gig*^RNAi^), which blocks Sestrin-mediated TORC1 inhibition [23] (Figure 5A). Similarly, *dSesn* expression rescues climbing speed in flies also expressing muscle-specific RNAi against *d4eBP* (*Mef2 > dSesn^WT^;Mef2 > Thor*^RNAi^), which uncouples TORC1 signaling and translational regulation [40] (Figure 5B). Fat body Lysotracker staining was low in flies expressing muscle-specific RNAi against either *TSC2* (Figure 5C) or *d4eBP* (Figure 5D), but wild-type *dSesn* expression rescued fat body Lysotracker (Figure 5C,D) without increasing Lysotracker in the myocardium (Figure 5E,F). These results indicate that inhibition of TORC1 is dispensable for *dSesn*’s impact on climbing speed and Lysotracker.

In order to further examine the role of TORC2 in this context, we expressed muscle-specific RNAi against the TORC2 complex protein-encoding *Sin1* in flies also expressing wild-type *dSesn* in muscles [30]. Muscle-specific *dSesn* expression could no longer improve climbing speed when *Sin1* was knocked down (Figure 6A). Similarly, neither exercise, nor *dSesn* overexpression could increase adipose or myocardial Lysotracker when *Sin1* was knocked down (Figure 6B–D). These results underscore the importance of the TORC2-AKT axis in the mobility and lysosomal adaptations conferred by Sestrin upregulation.

### 3.6. PGC1*α* Is Essential for the Beneficial Effects of Sestrin

We have previously shown that the transcriptional cofactor PGC1α is upregulated during Sestrin overexpression, and that *dSesn*-deficient *Drosophila* have reduced d*PGC1*α (*spargel*, *srl*) expression [30]. Likewise, d*PGC1*α was critical for the endurance and flight performance extending effects of Sestrin [30], highlighting *dPGC1**α* as an important downstream factor in the exercise-Sestrin pathway. Here, *dSesn* expression failed to improve climbing speed in flies expressing muscle specific RNAi against d*PGC1*α (*Mef2 > dSesn^WT^;Mef2 > srl*^RNAi^), (Figure 7A). d*PGC1*α RNAi also prevented exercise from inducing fat body lysosomal activity, even in the presence of *dSesn* overexpression (Figure 7B, upper panels in D) while myocardial Lysotracker staining was unaffected (Figure 7C,D). These results indicate that dPGC1α is critical for the effects of *dSesn* on speed and tissue-non-autonomous lysosomal activity.

## 4. Discussion

We have previously shown that Sestrin is required for exercise adaptations to endurance and metabolism in flies and mice, and that *dSesn* overexpression is sufficient to mimic physiological improvements to endurance and flight performance in *Drosophila*, even without training. Our previous work also implicated Akt and PGC1α as critical factors underlying the healthspan extending effects of Sestrin in flies and mice. Here, we examine whether *dSesn* plays a similar role in increasing speed and lysosomal activity in response to chronic exercise in *Drosophila*.

Both speed and endurance decline in aging individuals, and each contributes to reduced quality of life in distinct ways [41]. Better endurance is correlated with increased independence and enhanced executive function [42], while low gait speed is a significant predictor of frailty and cognitive decline [43,44]. Furthermore, declines in gait speed are usually accompanied by cardiac and metabolic dysfunction [41]. Autophagy is one conserved process by which cells exert protein and organelle quality control, and its precise regulation ensures proper cardiac and metabolic function during aging [45,46,47]. Identifying conserved factors capable of preserving these functions in animal models has great potential to generate novel therapeutic options for the aging population.

Sestrin, an evolutionarily conserved stress-inducible protein activated by endurance exercise, inhibits TORC1 while simultaneously activating TORC2 and inducing a wave of antioxidant protection [23,28,30]. mTOR is a nutrient-responsive protein kinase that regulates cell growth and metabolism in response to nutrient availability [48,49,50,51]. By activating protein and lipid anabolism and suppressing autophagic catabolism, mTOR positively regulates organ growth including muscles [49,52]. However, chronic upregulation of mTOR activity can induce excessive protein and lipid synthesis as well as chronic downregulation of autophagy [50,51,53,54,55], which can contribute to several pathologies including cancer, obesity and cardiac/skeletal muscle degeneration [53,54,55]. Over-nutrition and lack of exercise, also cause chronic mTOR activation, which can account for various age-associated pathologies, such as type II diabetes and cardiovascular diseases [50,51]. Conversely, exercise-induced energy depletion can inhibit mTOR through AMP-activated protein kinase (AMPK), attenuating age-associated pathologies and prolonging life and health span. Therefore, proper regulation of mTOR, especially the prevention of its chronic activation, is a key factor for prevention of age-associated diseases and improved quality of life in later ages. Here, we provide further support for Sestrin as a potent exercise mimetic that integrates catabolic effects from downregulation of TORC1 with maintenance of muscle and mitochondrial health driven by upregulation of Akt and PGC1α (Figure 8).

While some of the effects of *dSesn* on climbing speed parallel previous observations of *Drosophila* endurance, there are some important differences. For example, while AKT was essential to extend endurance and flight performance, our earlier study did not examine the impact of AMPKαon Sestrin’s mobility-enhancing effect. Sestrin inhibits TORC1 through AMPKα [27], and depletion of AMPKα in Sestrin-expressing flies only partially inhibits improvements to climbing speed. Interestingly, both Sestrin expression and exercise restore increases in lysosomal activity in AMPKα deficient flies, suggesting that Sestrin activation can increase lysosomal activity through pathways that do not require AMPK.

While this study clearly indicates that Sestrin is required for exercise to increase speed, just as we previously showed for endurance, there were some new wrinkles in the way *dSesn* interacts with exercise to increase Lysotracker staining. Unlike speed and endurance, *dSesn* is not absolutely required in any tissue for exercise to increase Lysotracker staining in the fat body. However, overexpression of *dSesn* in muscle can increase Lysotracker staining non-autonomously in the fat body in parallel with exercise. This indicates that exercise can increase fat body lysosomal activity through pathways that do not require *dSesn*. A strong candidate for such a pathway would be through upregulation of octopamine (OA) secretion, which we have shown is required for exercise adaptations [33], and requires expression of β-adrenergic receptors in the fat body [19]. Since we observed here that *dSesn* can act additively with exercise, it will be interesting in future work to better delineate the interactions between *dSesn* and OA in driving exercise adaptations.

Here, we show that Sestrin is essential for the benefits to climbing speed in *Drosophila*, and that muscle specific *dSesn* expression replicates exercise-like improvements to aging mobility. We also found that Sestrin mediates changes to lysosomal activity in multiple tissues, and that both of these adaptations depend on TORC2-Akt activity and on PGC1α. These findings reinforce the conclusion that Sestrins are critical factors modifying the physiological and subcellular benefits of endurance exercise across multiple animal models.

## Figures and Tables

**Figure 1 cells-10-02479-f001:**
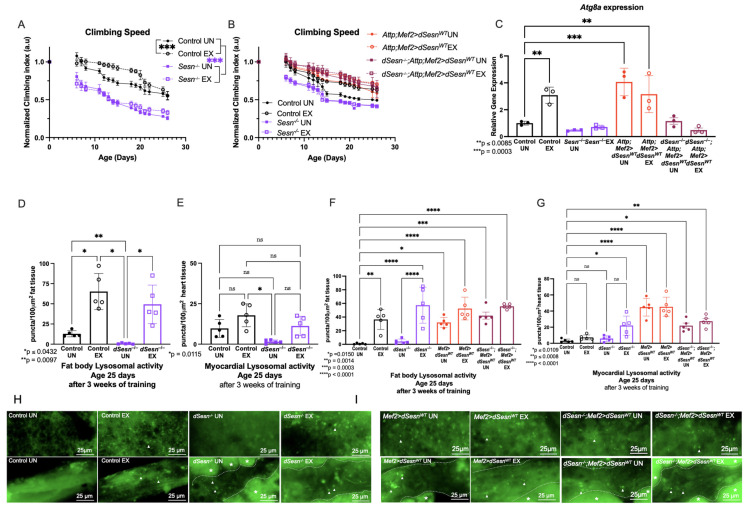
Sestrin drives exercise adaptations to climbing speed and lysosomal activity. (**A**) *dSesn^−/−^* flies have lower climbing speed than control flies whether exercised or not (2-way ANOVA, genotype effect, *p* ≤ 0.019 beginning at day 4). Exercise increases climbing speed in control flies (2-way ANOVA, exercise effect, *p* ≤ 0.0465, days 8–22). (**B**) Flies expressing *dSesn* in muscles in either a wild-type or *dSesn* null background (*Attp;Mef2 > dSesn^WT^, dSesn^−/−^; Attp;Mef2>dSesn^WT^,* respectively) have high climbing speed whether exercised or not (2-way ANOVA, genotype effect, *p* ≥ 0.1032). Climbing speed experiments performed in triplicate, n ≥ 100 flies. (**C**) *Atg8* gene expression is elevated in exercised background controls (ANOVA, *p* = 0.0085) and *Attp;Mef2 > dSesn^WT^* flies, whether exercised or not (ANOVA, *p* = 0.0003, EX *p* = 0.0064, UN). Data are presented as ΔΔCT normalized to *Atg8a* gene expression in unexercised Control flies. N = 20 flies per cDNA extraction, performed in triplicate. (**D**) Exercise increases fat body Lysotracker in control flies (ANOVA, *p* = 0.0284). Unexercised *dSesn^−/−^* flies have low fat body Lysotracker (ANOVA, *p* = 0.0097) but staining increases with exercise (ANOVA, *p* = 0.0432). (**E**) Myocardial Lysotracker staining is low in unexercised *dSesn^−/−^* flies (ANOVA, *p* = 0.0115). All other genotypes have similar myocardial Lysotracker staining. (**F**) Exercised *dSesn^−/−^; Attp;Mef2 > dSesn^WT^* flies have high fat body Lysotracker whether exercised or not (ANOVA, *p* ≤ 0.0014). (**G**) Myocardial Lysotracker staining is high in both *Attp;Mef2 > dSesn^WT^* and *dSesn^−/−^; Attp;Mef2 > dSesn^WT^* flies whether exercised or not (ANOVA, *p* ≤ 0.0109). (**H**) Representative 63× images of Lysotracker staining in fat body (upper panels) and myocardium (lower panels) from control and *dSesn^−/−^* flies, quantified in (**D**,**E**). (**I**) Representative 63× images of Lysotracker staining in fat body (upper panels) and myocardium (lower panels) from *Attp;Mef2 > dSesn^WT^* and *dSesn^−/−^; Attp;Mef2 > dSesn^WT^* flies, quantified in (**F**,**G**). Lysotracker staining n = 5, performed in duplicate or triplicate. Dotted lines in myocardial images delineate heart tissue. White triangles point to example regions of puncta. White asterisks indicate pericardial cells. Brackets represent pairwise post-hoc comparisons from ANOVA. ns or lack of brackets indicates no significant difference. Error bars depict ± SD. EX = cohort after 3 weeks of ramped, daily endurance training. UN = control unexercised flies exposed to identical training environment without running.

**Figure 2 cells-10-02479-f002:**
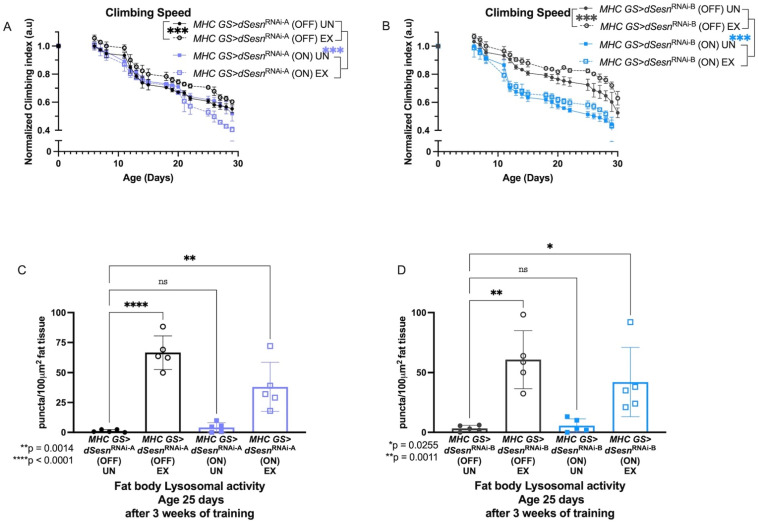
Sestrin RNAi in adult muscles abrogates exercise-dependent increases in climbing speed and lysosomal activity. (**A**) RNAi against *Sestrin* in adult muscles (*MHC GS > dSesn^RNA-A^* (ON) UN) reduces climbing speed (2-way ANOVA, *p* ≥ 0.0489), and exercise further reduces climbing speed, particularly at later timepoints (2-way ANOVA, *p* ≤ 0.0184 after adult day 22). Control flies respond to exercise as normal. (**B**) A separate RNAi construct against *Sestrin* in adult muscles (*MHC GS > dSesn^RNA-B^* (ON) UN) also reduces climbing speed (2-way ANOVA, genotype effect, *p* ≤ 0.0046 after adult day 8). Control flies respond to exercise as normal. Climbing experiments performed in duplicate, n ≥ 100 flies. (**C**) RNAi against *dSesn* in muscle reduces exercise-induced increase in lysosomal activity in fat body (ANOVA, *p* = 0.0014), but *MHC GS > dSesn^RNA-A^* (ON) flies still increase fat body lysosomal activity compared to unexercised siblings (ANOVA, *p* = 0.0030). (**D**) *MHC GS > dSesn^RNA-B^* (ON) flies also increase fat body Lysotracker staining after exercise (ANOVA, *p* = 0.0372). EX = cohort after 3 weeks of ramped, daily endurance training. UN = control unexercised flies exposed to identical training environment without running. ns or lack of brackets indicates no significant difference.

**Figure 3 cells-10-02479-f003:**
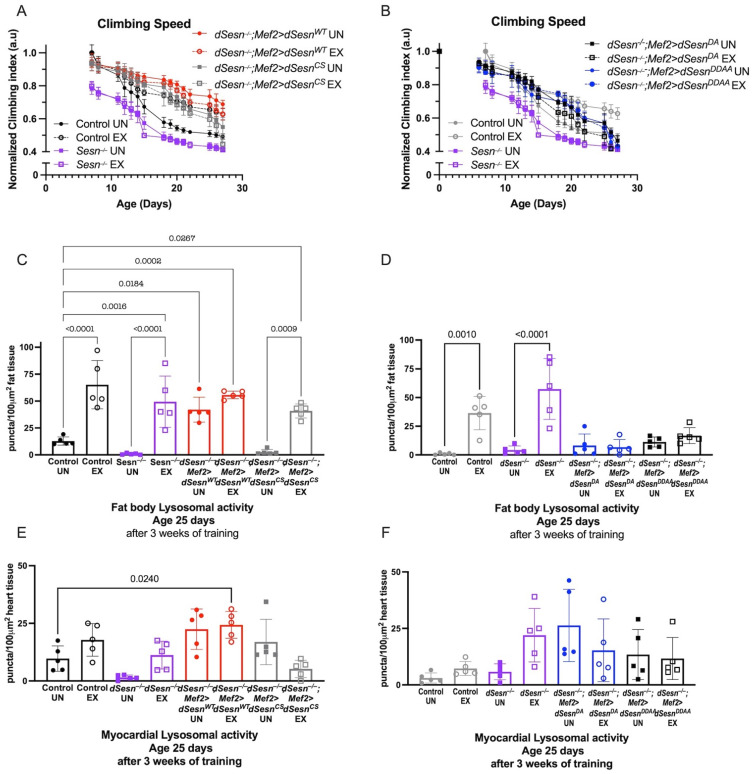
Wild-type *dSesn* expressed in muscle of *dSesn*^−/−^ flies improves climbing speed and increases fat body lysosomal activity. Wild-type, C86S, D423A, D423A/D424A *dSesn* transgenes were expressed in muscle of *dSesn*^−/−^ flies. (**A**,**B**) *dSesn^−/−^;Mef2 > dSesn^WT^* climb faster than *dSesn^−/−^;Mef2 > dSesn^CS^* (shown in (**A**)), *dSesn^−/−^;Mef2 > dSesn^DA^* and *dSesn^−/−^;Mef2 > dSesn^DDAA^* flies (shown in (**B**)) whether exercised or not (2-way ANOVA, genotype effect, *dSesn^−/−^;Mef2 > dSesn^WT^* compared to unexercised and exercised *Mef2 > dSesn^CS^, dSesn^−/−^;Mef2 > dSesn^DA^* and *dSesn^−/−^;Mef2>dSesn^DDAA^* f lies: *p* ≤ 0.0335, d15+). Climbing speed experiments performed in triplicate, n ≥ 100 flies. (**C**,**D**) *dSesn^−/−^;Mef2>dSesn^WT^* flies have higher Lysotracker staining in fat bodies than *dSesn^−/−^;Mef2 > dSesn^CS^* (shown in (**C**), compare to red bars), *dSesn^−/−^;Mef2 > dSesn^DA^* and *dSesn^−/−^;Mef2 > dSesn^DDAA^* flies (shown in (**D**), compare to red bars in (**C**)) (ANOVA, *p* ≤ 0.0319). Exercise increases fat body Lysotracker in *dSesn^−/−^;Mef2 > dSesn^CS^* (shown in **C**) (ANOVA, *p* < 0.0001) (**E**) Exercised *dSesn^−/−^;Mef2 > dSesn^CS^*flies have lower myocardial Lysotracker than *dSesn^−/−^;Mef2 > dSesn^WT^* flies (ANOVA, *p* ≤ 0.0102). (**F**) Myocardial Lysotracker staining in *dSesn^−/−^;Mef2 > dSesn^DA^* and *dSesn^−/−^;Mef2 > dSesn^DDAA^* flies is similar to flies expressing Wild-type *dSesn* (Compare to red bars in (**E**)) whether exercised or not (ANOVA, *p* ≥ 0.3075). Lysotracker staining n = 5, performed in duplicate or triplicate. Brackets represent pairwise post-hoc comparisons from ANOVA. Error bars depict ± SD. EX = cohort after 3 weeks of ramped, daily endurance training. UN = control unexercised flies exposed to identical training environment without running. ns or lack of brackets indicates no significant difference.

**Figure 4 cells-10-02479-f004:**
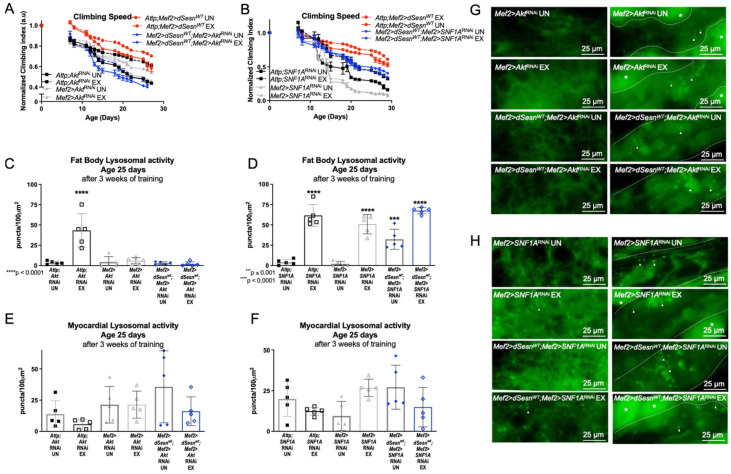
AKT is required for Sestrin and exercise to improve climbing speed and enhance lysosomal activity. (**A**) Flies overexpressing *dSesn* in muscles (*Attp;Mef2 > dSesn^WT^*) climb faster than controls and climb as fast as exercised controls (2-way ANOVA, *p* ≤ 0.0156, *p* ≥ 0.1300). Exercised flies expressing RNAi against AKT in muscles (*Mef2 > Akt*^RNAi^ UN) improve climbing speed but do not reach the speed of exercised controls (2-way ANOVA, *p* ≤ 0.0319). *dSesn* expression does not rescue climbing speed (*Mef2 > dSesn^WT^; Mef2 > Akt*^RNAi^) (2-way ANOVA, genotype effect, *Mef2 > Akt*^RNAi^ UN vs. *Mef2 > dSesn^WT^; Mef2 > Akt*^RNAi^ UN, *p* ≤ 0.0463). (**B**) Flies expressing RNAi against AMPKα in muscles (*Mef2 > SNF1A*^RNAi^) have low climbing speed and do not improve with exercise (2-way ANOVA, genotype effect, *p* ≤ 0.0070 after day 12, exercise effect, *p* ≥ 0.4340). *dSesn* expression does not rescue climbing speed to the level of *Attp;Mef2 > dSesn^WT^* flies in flies with muscle-specific AMPKα RNAi (*Mef2 > dSesn^WT^*;*Mef2 > SNF1A*^RNAi^) (2-way ANOVA *p* ≤ 0.0269 after day 12). Climbing speed experiments performed in duplicate or triplicate, n ≥ 100 flies. (**C**) Fat body Lysotracker staining is low in both *Mef2 > Akt*^RNAi^ and *Mef2 > dSesn^WT^; Mef2 > Akt*^RNAi^ flies whether exercised or not (compare to *Attp;Akt*^RNAi^ EX, ANOVA, *p* < 0.0001). (**D**) Exercised *Attp;SNF1A*^RNAi^, *Mef2 > SNF1A*^RNAi^, and *Mef2 > dSesn^WT^*;*Mef2 > SNF1A*^RNAi^ flies increase fat body Lysotracker staining with exercise (ANOVA, *p* < 0.0001). (**E**,**F**) Myocardial Lysotracker staining is similar in background controls and *Mef2 > Akt*^RNAi^ and *Mef2 > SNF1A*^RNAi^ flies independent of both wild-type *dSesn* overexpression and exercise. Representative 63× images of Lysotracker staining in fat body (left panels) and myocardium (right panels) of exercised and control flies expressing muscle-specific RNAi against (**G**) *Akt* and (**H**) AMPKα, with and without muscle-specific *dSesn* expression. Lysotracker staining n = 5, performed in duplicate or triplicate. Dotted lines in myocardial images delineate heart tissue. White triangles point to example regions of Lysotracker puncta. White asterisks indicate pericardial cells. Asterisks in histograms represent significant post-hoc comparisons from ANOVA. Error bars depict ± SD. EX = cohort after 3 weeks of ramped, daily endurance training. UN = control unexercised flies exposed to identical training environment without running.

**Figure 5 cells-10-02479-f005:**
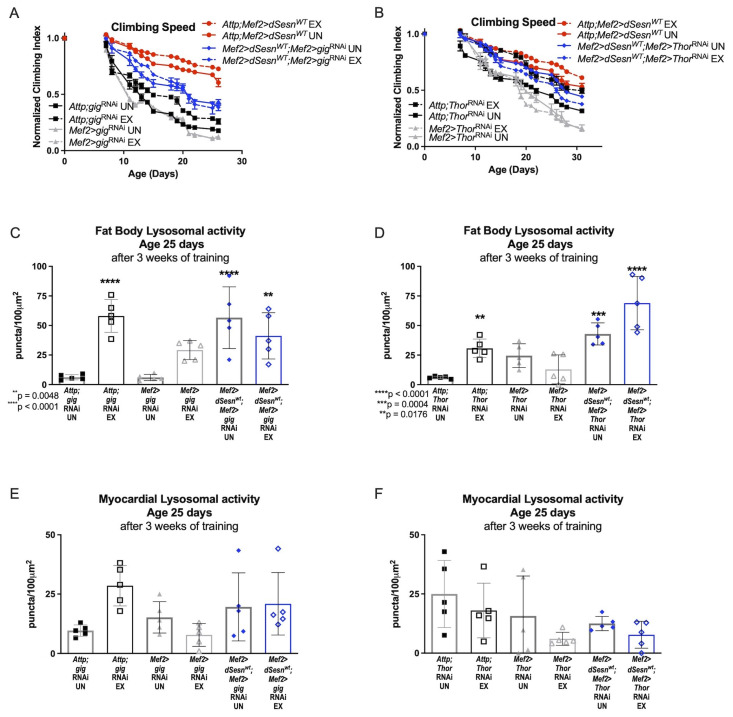
Neither TSC-2 nor 4eBP are required for climbing speed and lysosomal adaptations conferred by Sestrin. (**A**) Exercised flies expressing muscle specific RNAi against TSC-2 (*Mef2 > gig*^RNAi)^ do not improve climbing speed (2-way ANOVA, exercise effect, *p* ≥ 0.6037, but *dSesn* expression (*Mef2 > dSesn^WT^*;*Mef2 > gig*^RNAi^ flies) rescues climbing speed (2-way ANOVA, *p* ≤ 0.0283). (**B**) Flies expressing muscle specific RNAi against 4eBP (*Mef2 > Thor*^RNAi^ flies) have low climbing speed and do not improve climbing speed with exercise (2-way ANOVA, exercise effect, *p* ≥ 0.8797, genotype effect, *p* ≤ 0.0219). Wild-type *dSesn* expression (*Mef2 > dSesn^WT^*;*Mef2 > Thor*^RNAi^ flies) rescues climbing speed (2-way ANOVA, *p* ≥ 0.2350). Climbing speed experiments performed in triplicate, n ≥ 100 flies. (**C**,**D**) Fat body Lysotracker staining is higher than controls in both *Mef2 > dSesn^WT^*;*Mef2 > Thor*^RNAi^ and *Mef2 > dSesn^WT^*;*Mef2 > gig*^RNAi^ whether exercised or not (ANOVA, *p* ≤ 0.0048). (**E**,**F**) Myocardial Lysotracker staining is similar across genotypes independent of exercise and wild-type *dSesn* expression in flies with muscle specific RNAi against either TSC-2 or 4eBP. Lysotracker staining n = 5, performed in duplicate or triplicate. Asterisks represent significant post-hoc comparisons from ANOVA. Error bars depict ± SD. EX = cohort after 3 weeks of ramped, daily endurance training. UN = control unexercised flies exposed to identical training environment without running.

**Figure 6 cells-10-02479-f006:**
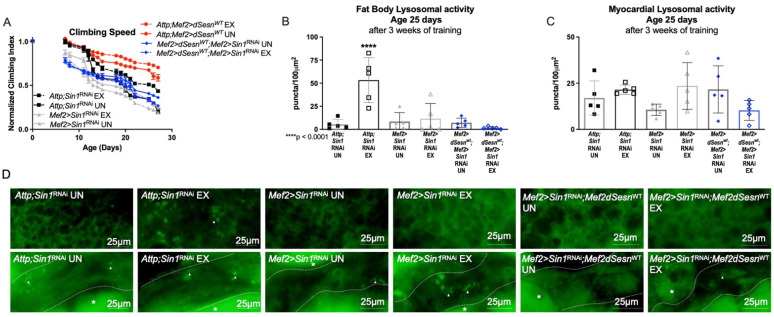
Sestrin-dependent adaptations in climbing speed and lysosomal activity depend on the TORC2 axis. (**A**) Flies expressing muscle specific RNAi against *Sin1* (*Mef2 > Sin1*^RNAi)^ reduce climbing speed with exercise (2-way ANOVA, exercise effect, *p* ≥ 0.0268). Muscle-specific wild-type *dSesn* expression (*Mef2 > dSesn^WT^*;*Mef2 > Sin1*^RNAi^ flies) does not rescue climbing speed. Climbing speed experiments performed in triplicate, n ≥ 100 flies. (**B**) *Mef2 > dSesn^WT^*;*Mef2 > Sin1*^RNAi^ flies have low fat body Lysotracker staining whether exercised or not. (**C**) Myocardial Lysotracker staining is similar across genotypes independent of exercise and wild-type *dSesn* expression in flies with muscle specific RNAi against *Sin1*. (**D**) Representative 63× images of Lysotracker staining from fat body (upper panels, quantified in (**B**)) and myocardium (lower panels, quantified in (**C**)). Lysotracker staining n = 5, performed in duplicate or triplicate. Dotted lines in myocardial images delineate heart tissue. White triangles point to example regions of Lysotracker puncta. White asterisks indicate pericardial cells. Asterisks in histograms indicate significant pairwise post-hoc comparisons from ANOVA. Error bars depict ± SD. EX = cohort after 3 weeks of ramped, daily endurance training. UN = control unexercised flies exposed to identical training environment without running.

**Figure 7 cells-10-02479-f007:**
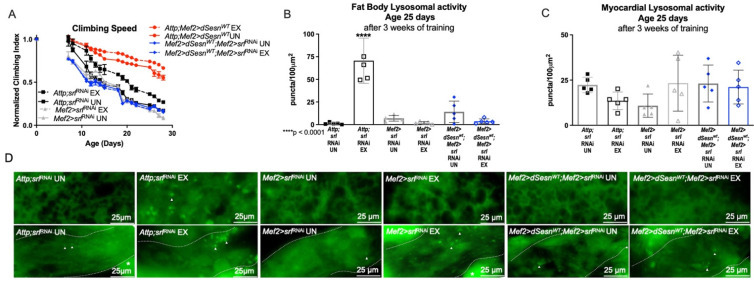
PGC1α is required for exercise- and Sestrin-dependent adaptations to climbing speed and lysosomal activity. (**A**) Flies expressing RNAi against PGC1α in muscles have a climbing speed similar to unexercised background controls independent of both *dSesn* expression and exercise (*Mef2 > srl*^RNAi^ flies, *Mef2 > dSesn^WT^*;*Mef2 > srl*^RNAi^ flies, 2-way ANOVA, *p* ≥ 0.2168). Climbing speed experiments performed in duplicate, n ≥ 100 flies. (**B**) *Mef2 > dSesn^WT^*;*Mef2 > srl*^RNAi^ flies have low fat body Lysotracker staining whether exercised or not. (**C**) Myocardial Lysotracker staining is similar across genotypes independent of exercise and wild-type *dSesn* expression. (**D**) Representative 63× images of Lysotracker staining in fat body (upper panels) and myocardium (lower panels) of exercised and control flies expressing muscle-specific RNAi against PGC1α. Lysotracker staining n = 5, performed in triplicate. Dotted lines in myocardial images delineate heart tissue. White triangles point to example regions of Lysotracker puncta. White asterisks indicate pericardial cells. Error bars depict ± SD. EX = cohort after 3 weeks of ramped, daily endurance training. UN = control unexercised flies exposed to an identical training environment without running.

**Figure 8 cells-10-02479-f008:**
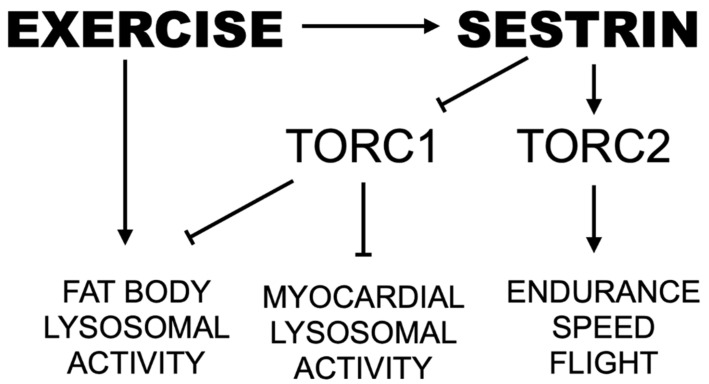
Proposed pathway by which endurance exercise and Sestrin mediate speed and lysosomal activity. Arrows indicate positive regulation. Lines with perpendicular end caps indicate negative regulation.

## Data Availability

The datasets generated and analyzed in this study are available from the corresponding authors on request. The source data underlying Figure 1A–C,F,I, Figure 2 and Figure 3, Figure 4A–F, Figure 5A–C, Figure 6A–C and Figure 7A–C are provided in Appendix A as a Source Data File.

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
