# Peer review of "Exercise and Sestrin Mediate Speed and Lysosomal Activity in Drosophila by Partially Overlapping Mechanisms"

_cells, 2021, doi:10.3390/cells10092479_

Round 1

Reviewer 1 Report

This manuscript focuses on the role of exercise, sestrin and various signal transduction pathways on Drosophila climbing speed and lysosome accumulation. In general, the conclusions appear to be sound. Overall, this is a nice paper, but the large number of genotypes and comparisons make it difficult to digest. The authors do a reasonable job of summarizing the key points, however.  There are some concerns about the text and figures that should be considered by the authors in revisions.

Line 50: The segue from the paragraph above to this one about Sestrins is poor. Adding a sentence to the previous paragraph such as “One such group of factors is the Sestrins” would be helpful to ease the transition.

Lines 74-76: dSesn has additive effects with exercise to induce lysosomal activity, but not in adipose tissue. Need to clarify what tissue the first part of the sentence refers to.

Figs 1D,E: no comparison of dSesn-/- UN to EX? Fig 1E: Control UN to EX comparison? Or does lack of comparison mean no statistical difference? If so, what does the ns comparison in D indicate?

Fig 1 Legend. Define UN and EX here and elsewhere.

Fig 1 H/I. Aside from the left-most panel it is unclear where the heart tube is within the images. Perhaps dotted lines need to be drawn around the tissue area being assessed? Also, for right panels in Fig 4G/H, Fig 6D, Fig 7D.

For all of the Lysotracker images, it is difficult to know what the authors are quantifying as puncta. Some images have clear puncta. Others have very bright staining with few puncta.

Fig 3A, what is being compared to give the *** significance? The difference between each genotype for UN and EX? This needs to be more explicitly stated.

For Fig 3D, it is unclear what is being compared. Presumably values are compared to the red bars in Figure C? Perhaps the two graphs should be merged? Similarly for E and F?

Line 254: Could the partial effect of AMPKalpha knockdown arise from partial knockdown by RNAi?

Fig 4A. There is an unconnected blue point at the lower right of the graph.

Fig. 5B legend: how does the p value of 0.2350 show rescue of climbing speed? Are you comparing to the rescued value?

Author Response

We thank the reviewers for their thoughtful suggestions.  Please see attachment.  Responses are highlighted in red and direct changes to text have been copied directly into our response.

Reviewer 2 Report

In the study titled with “Exercise and Sestrin mediate speed and lysosomal activity in Drosophila by partially overlapping mechanisms” by Alyson Sujkowski et al. reported the roles of dSestrin in preserving climbing speed and increasing lysosomal activity. The author applied a well-designed exercise training system and found that the regulation of dSestrin in exercise adaptation require other factors, including TORC2/AKT axis and PGC1⍺, while the interaction with TORC1 is required for cell-autonomously increase of lysosome activity. The manuscript is well written, and I would support this paper to be published. Before that, a few questions or concerns need to be answered or demonstrated properly.

1)There are some cases that control data were missing in figures which make the data in the graph not easy to evaluate:

The wild-type and dSesn-/- mutant climbing index data missing in figure 1B, could the author make this data included so that we could clearly see the increase of climbing speed in muscle specific dSesn overexpression flies?

Please include the Atg8 expression data from dSesn-/-; Attp; Mef2>dSesnWT flies in figure 1C.

Please include the Lysotracker data of dSesn-/- mutant flies in figure 1F and 1G.

Please include the wild-type and dSesn-/- mutant climbing index data and lysotracker data in figure 3(figure 3A-F).

2)In line 219, the author said that dSesn acts in muscle to cell-autonomously drive Lysotracker staining in myocardium. I’d like to see the lysotracker data of myocardium in dSesn muscle specific KD flies. The author could put the data in the supplementary information.

3)Does the author have any explanation that in figure 3A, the unexercised flies (UN) retain a faster climbing speed than age-matched exercised siblings (EX) or is it a mislabeling?

4)Chronic exercise is recognized as an important contributor to health span in animal, and the data from this manuscript suggests that sestrin is essential for the benefits to climbing speed. It will be interesting to see whether these exercised flies live longer or not and whether sestrin is also required. And what strength/ intensity of exercise is the best for a longer life span? Did the author notice any related studies? It could be included in discussion or future directions.

Minor Points:

1)Line 18, missing full stop, “increase lysosomal activity These results” should be” increase lysosomal activity. These results”

2)Line 300, “chronic exercise in Drosophila. sestrin” should be “chronic exercise in Drosophila.”

Author Response

(The authors gave the same response as above.)
